# Tumor Microenvironment and Hydrogel-Based 3D Cancer Models for In Vitro Testing Immunotherapies

**DOI:** 10.3390/cancers14041013

**Published:** 2022-02-17

**Authors:** Chiara Vitale, Monica Marzagalli, Silvia Scaglione, Alessandra Dondero, Cristina Bottino, Roberta Castriconi

**Affiliations:** 1Department of Experimental Medicine (DIMES), University of Genova, 16132 Genova, Italy; chiara.vitale@edu.unige.it (C.V.); alessandra.dondero@unige.it (A.D.); roberta.castriconi@unige.it (R.C.); 2React4life SRL, 16121 Genova, Italy; m.marzagalli@react4life.com (M.M.); s.scaglione@react4life.com (S.S.); 3National Research Council of Italy, Institute of Electronics, Information Engineering and Telecommunications (IEIIT), 16149 Genova, Italy; 4IRCCS Istituto Giannina Gaslini, 16147 Genova, Italy

**Keywords:** 3D cancer models, immunotherapies, biomaterials, tumor escape mechanisms

## Abstract

**Simple Summary:**

Immunotherapies are emerging as promising strategies to cure cancer and extend patients’ survival. Efforts should be focused, however, on the development of preclinical tools better able to predict the therapeutic benefits in individual patients. In this context, the availability of reliable preclinical models capable of recapitulating the tumor milieu while overcoming the limitations of traditional systems is mandatory. Here, we review the tumor immune responses, escape mechanisms, and the most recent 3D biomaterial-based cancer in vitro models useful for investigating the effects of the different immunotherapeutic approaches. The main challenges and possible future trends are also discussed.

**Abstract:**

In recent years, immunotherapy has emerged as a promising novel therapeutic strategy for cancer treatment. In a relevant percentage of patients, however, clinical benefits are lower than expected, pushing researchers to deeply analyze the immune responses against tumors and find more reliable and efficient tools to predict the individual response to therapy. Novel tissue engineering strategies can be adopted to realize in vitro fully humanized matrix-based models, as a compromise between standard two-dimensional (2D) cell cultures and animal tests, which are costly and hardly usable in personalized medicine. In this review, we describe the main mechanisms allowing cancer cells to escape the immune surveillance, which may play a significant role in the failure of immunotherapies. In particular, we discuss the role of the tumor microenvironment (TME) in the establishment of a milieu that greatly favors cancer malignant progression and impact on the interactions with immune cells. Then, we present an overview of the recent in vitro engineered preclinical three-dimensional (3D) models that have been adopted to resemble the interplays between cancer and immune cells and for testing current therapies and immunotherapeutic approaches. Specifically, we focus on 3D hydrogel-based tools based on different types of polymers, discussing the suitability of each of them in reproducing the TME key features based on their intrinsic or tunable characteristics. Finally, we introduce the possibility to combine the 3D models with technological fluid dynamics platforms, reproducing the dynamic complex interactions between tumor cells and immune effectors migrated in situ via the systemic circulation, pointing out the challenges that still have to be overcome for setting more predictive preclinical assays.

## 1. Introduction

During the last decade, immunotherapy has emerged as a promising alternative to traditional anticancer treatments [1,2,3,4]. Harnessing the immune system represents a potent approach providing a patient-specific and durable strategy. The benefit and efficacy of immunotherapies, unleashing immune cell activity through antibodies specific for tumor antigens (TAs) or blocking immune checkpoint axes (ICB), have been documented in several clinical trials. However, in a significant portion of patients, resistance to immunotherapeutic procedures exists, whose causes are yet to be clarified. Indeed, immunotherapies, established according to results obtained from current preclinical models, resulted in only 20–40% of durable clinical responses [5]. In light of these considerations, there is an urgency to validate new platforms suitable to better set up anticancer treatments and predict their efficacy in individual patients. To this aim, it is imperative to develop highly predictive screening tools capable of resembling the complex structure of the human tumor microenvironment (TME), holding a heterogeneous population of cells and extracellular components, all involved in dynamic crosstalk. Then, ideally, such platforms might consider both the 3D contexture of the tissue and the complex interplay between different cell types.

Several cancer-related phenomena, such as metastasis, cell motility, and uncontrolled proliferation, are modulated by the surrounding extracellular matrix (ECM) [6]. Indeed, the ECM provides both mechanical support and biochemical signals, directly affecting cell activity in both physiological and pathological conditions [7]. In a solid tumor scenario, cancer cells are capable of shaping the ECM niche by changing its properties toward a pro-malignant phenotype [8]. Experimental evidence such as the histological analysis of tumor specimens isolated from animal models or patients strongly supports the concept that an altered ECM architecture and composition can play a pivotal role in the regulation of tumor onset, progression, immune evasion, and sensitivity to immunotherapeutic approaches [9,10]. ECM components, including glycoproteins, glycosaminoglycans (GAG), proteoglycans, soluble molecules, and fibrillar proteins, can directly interact with specific receptors on the tumor cell surface, regulating several aspects of their biology. In this context, particularly relevant is the CD44/hyaluronic acid (HA) axis [11], which is targeted in different therapeutic approaches, or the CXCR4/SDF1 axis driving bone marrow (BM) homing of tumor cells. ECM components can also contribute to the onset of an immunosuppressive milieu since many molecules such as the serine protease plasmin, matrix metalloprotease (MMP)-2 and MMP9, or thrombospondin-1 disrupt the latent form of tumor growth factor (TGF)-β (LAP–TGF-β) with the release of an active form of the cytokine [12,13]. Upon activation, TGF-β exerts potent immunomodulatory functions shaping the tumor immune landscape (see next paragraphs for more details). The ECM, depending on the relative abundance of the various components, can form “molecular sieves” able to regulate the migration of tumor cells and immune cells inside the ECM. In this context, it is relevant to stress that in tumor specimens from patients, components of the immune system such as natural killer (NK) cells are mainly observed in the tumor stroma rather than in tumor parenchyma [14,15], probably contributing to the low efficacy of the NK-mediated immune surveillance of solid tumors. Low tumor infiltration is also observed in the context of adoptive cell therapy based on the infusion of T cells, in vitro engineered to expressed TA-specific chimeric receptors (CAR-T) [16]. These effectors were shown to lose the expression of the enzyme heparanase (HPSE), which degrades heparan sulfate proteoglycans, the main components of the ECM [16]. Importantly, when CAR-T cells were also transduced with HPSE, they showed enhanced tumor infiltration and improved overall survival in xenograft tumor models. It is relevant to point out that the architecture of the extracellular fibers such as collagen also greatly influences the tumor biology and response to therapies [17]. In particular, it has been recently shown that the interaction of discoidin domain receptor 1 (DDR1), a collagen receptor expressed by tumor cells, promotes collagen fiber alignment, contributing to immune exclusion [18]. Ablation of DDR1 in tumors promotes the intra-tumoral infiltration of T cells. The described collagen remodeling requires the DDR1 extracellular (DDR1-ECD), but not the intracellular, kinase domain to be effective. Importantly, fibrillar components also impact the status of tumor cells in terms of quiescence or proliferation. In this context, “very dormant” cancer cells contribute to the establishment of type III collagen-enriched ECM niches that deeply sustain tumor dormancy. Histopathological analysis showed that tumor specimens from patients with lymph node-negative head and neck squamous cell carcinoma were enriched in type III collagen levels as compared to tumors from patients with tumor-infiltrated lymph nodes [19].

Besides the ECM components, tumor-infiltrating cells are key determinants of malignant advancement. Different cell types including regulatory T cells (Treg), cancer-associated fibroblasts (CAFs), tumor-associated macrophages (TAMs), mesenchymal stem cells, and endothelial cells can all contribute to tumor growth and escape from the host immune surveillance [20,21,22]. Therefore, to closely mimic in vitro the complex cancer dynamic environment, it is crucial to consider the tumor–stroma–immune cell interplays.

To date, most of the data available on the communication between the human immune system and cancer cells rely on 2D standard monolayers [23]. These flat systems are standardized, high-throughput, and cost-effective [24]. However, they are over-simplified tools that cannot replicate the complexity of the in vivo scenario, mainly due to the bi-dimensionality resulting in the lack of proper cell-to-cell and cell-to-ECM reciprocal interactions [23,25,26]. Moreover, although animal testing remains a gold standard in cancer research, it cannot faithfully reproduce the human TME [27]. In addition, animal models are costly, time-consuming, and hardly applicable on the road to large-scale personalized therapeutic approaches [28,29].

To solve these constraints, various 3D in vitro culture models have been realized to carry out preclinical experimental investigations under more physiological conditions. Initially, 3D multicellular tumor spheroids were proposed to imitate the human tumor native spatial arrangement, cellular reciprocal interplays, and diffusion gradients [30]. Such models revealed a closer resemblance to what occurs in vivo (cancer cell phenotype, proliferation rate, and drug resistance) when compared to 2D monolayers [31] and have allowed conducting systematic investigations in a reproducible manner that cannot be achieved with conventional models [32]. However, 3D in vitro culture models still have certain limitations, mainly due to the lack of an ECM favoring physical disintegration during their manipulation. Moreover, they do not allow obtaining information regarding cell interactions with the surrounding microenvironment. As a result, 3D scaffold-based cancer models have been more recently integrated with ECM components to better mimic pathophysiological features of native tumor tissues.

Merging the aforementioned considerations, here, we firstly describe the major tumor escape mechanisms and the therapeutic approaches potentiating the antitumor immune responses. Then, we review recent studies aimed at investigating tumor and immune system interactions and testing immunotherapeutic anticancer treatments by adopting 3D biomaterial-based cancer models. In particular, we focus on polymeric matrices used in the form of hydrogels. We present a schematic overview of the most important natural and synthetic biomaterials that have been adopted in this field, highlighting both benefits and limitations, and discussing how they can be optimized to fulfill some pivotal function of the tumor ECM.

Finally, we illustrate recent emerging microfluidic technologies that couple 3D hydrogel-based models with fluidical stimuli, thus mimicking the dynamic stimuli experienced by cells in vivo and affecting their physio-pathological behavior as well as their interplay with the immune system. Overall, these models could help to identify novel mechanisms making tumors resistant to immunotherapy, and to optimize innovative and personalized immunotherapeutic approaches, accelerating the clinical translation.

## 2. Mechanisms Allowing Cancer Immune Evasion, a Lesson from 2D Cultures and Animal Models

Several efforts have been made in recent years to modify the human TME to overcome the limitations related to the species-specific gaps existing in animal models. However, modeling the TME, from an immune point of view, is still challenging due to the highly complex relationships between cancer cells and immune cells. A plethora of immune cells can interact with tumor cells and the TME, and, depending on the nature of these interactions, immune effectors can acquire either a tumor-suppressive or tumor-promoting function. In addition, immune cells do not act alone but interact with each other, orchestrating tumor immune responses [33,34].

A huge amount of data indicates that a functional cancer immunosurveillance process exists. However, the relationship between cancer and immune cells is a complex dynamic process involving three phases, namely, Elimination, Equilibrium, and Escape, the so-called “3E’s of cancer immune-editing”. The Elimination phase is characterized by the successful activation of the immune system, leading to cancer cell recognition and death. In the Equilibrium phase, cancer cells adapt to the hostile environment established by the antitumor immune cells, enabling their survival and cohabitation. In the Escape phase, cancer cells, edited by the immune system, evade its aggression through mechanisms including the expression/upregulation of membrane-bound inhibitory axes, and the production of immunosuppressive soluble molecules [35]. The major tumor escape strategies are briefly described in the following paragraphs and illustrated in Figure 1.

### 2.1. Membrane-Bound Inhibitory Axes and Therapeutic Approaches

#### 2.1.1. HLA-I-Related Axes and Inhibitory Immune Checkpoints

Relevant membrane-bound inhibitory axes, negatively impacting the antitumor activity of innate and adaptive cytotoxic cells, are those involving HLA class I (HLA-I) molecules on target cells, and specific inhibitory receptors on effector cells, such as killer Ig-like receptors (KIRs), NKG2A, and LIR-1 [36]. These interactions have physiological functions: for example, licensing NK cells to acquire a suitable cytolytic potential [37,38] or tuning the activity of triggering receptors such as NKp46, NKp30, and NKp44 (collectively termed natural cytotoxicity receptors, NCRs), mainly expressed by NK cells, NKG2D, and DNAM-1, also characterizing a significant population of T cells [36,39]. HLA-I^high^ autologous healthy cells are generally protected from the NK cell-mediated aggression since the strength of the inhibitory signals prevails over that of the activating signals; the activating signals overcome the inhibitory signals in pathological conditions including tumors, where cell transformation leads to partial or complete HLA-I expression, together with the upregulation or de novo expression of ligands for activating receptors [36,39,40]. In some instances, however, tumors can preserve high levels of protective HLA-I, as occurs in hematological malignancies [41], or upregulate HLA-I as an adaptive mechanism to the IFN-γ and TNF-α mediators released by cytotoxic cells during tumor aggression [35,42].

To potentiate the cytotoxic antitumor responses, different strategies breaking the inhibitory receptor/HLA-I axes have been planned including those blocking KIRs [43] or NKG2A [44] with specific antibodies or hemopoietic stem cell (HSC) transplant with selected allogenic donors, generally the patient’s parents (haploidentical HSC, haplo-HSC) who can have NK cell populations with a KIR repertoire unable to recognize HLA-I alleles on the donors’ (KIR-KIR Ligand-mismatched NK cells). A further advance in the transplant setting is represented by TCR αβ/CD19-depleted haplo-HSC transplant, where cells infused in the recipient contain CD34+ HSC cells and mature immune cells including γδ T cells and NK cells, which provide early and effective antitumor and antiviral activities acting before the immune cell reconstitution from CD34+ cells [45].

The antitumor activity of cytotoxic cells can also be negatively regulated by several non-HLA-I-specific co-inhibitory receptors such as PD-1, LAG-3, and TIM-3, expressed by T and NK cells, interacting with ligands on tumor cells [46,47,48]. As with HLA-I, their ligands, PD-Ls (-L1 and -L2), HLA-II, and galectin-9, can be upregulated/induced by INF-γ released during the immune responses. Interestingly, an opposite regulation by INF-γ has been observed for PVR (poliovirus receptor, CD155) [23], a ligand shared by the inhibitory checkpoints TIGIT and CD96, and the activating DNAM-1 receptor.

The PD-1/PD-Ls and CTLA/CD28 axes, whose discovery was awarded with the 2018 Nobel Prize for Medicine and Physiology, represent the prototypic immune checkpoints firstly targeted in cancer patients. In particular, the blockade of the PD-1/PD-Ls axis has revolutionized the treatment of many metastatic advanced tumors either as monotherapy or in combination with other therapeutic strategies. Several clinical trials combine PD-1/PD-Ls blockade with the infusion of antibodies specific for Tas, which unleash cytotoxicity and IFN-γ release by NK and T cells, promote phagocytosis, and complement activation. The use of antibodies as bullets reaching the right target, sparing normal cells, represents a strategy commonly used in tumors characterized by a high expression of the selected antigen that, conversely, shows a limited/low expression in normal tissues. These antibodies are often engineered to be more effective, for example, by mutating their Fc portion to reduce their binding with inhibitory or low-affinity FcγRs, or conjugating them with toxic drugs [49]. Recently, to further improve the cytotoxicity of NK cells, a strategy has been developed based on multifunctional engagers simultaneously targeting Tas and CD16 (FcγRIIIA) and NKp46 activating receptors in NK cells [50].

Additional tools to efficiently and specifically target both hematological malignancies and solid tumors are represented by T cells engineered with chimeric antigen receptors (CARs) specific for Tas, which have been optimized in recent years with the construction of more effective third-generation CARs, which also express an inducible suicide gene to induce, in case of adverse side effects, the rapid in vivo depletion of CAR-T. CARs deliver a cell activation signal that, in most instances, is strong enough to overcome the inhibitory axes. Recently, there has been an increasing interest in the generation of CAR-engineered NK cells, effectors that appear to be superior in terms of safety and that naturally express different receptors against tumor-associated molecules [51,52].

#### 2.1.2. Novel Inhibitory Immune Checkpoints: B7-H3 and CD47

One of the most promising recently discovered tumor targets is B7-H3 (CD276) [53,54,55], highly expressed by several tumors and upregulated by IFN-γ [23]. Importantly, B7-H3 also shows a higher expression on the tumor-associated vasculature compared to normal vessels, whereas it is not expressed at significant levels on most normal tissues. B7-H3 represents an additional ligand of the growing list of immune checkpoint axes, physiologic mechanisms controlling the duration and the resolution of the immune responses [53,56,57,58]. Unfortunately, these inhibitory axes are “adopted” by tumors to escape immune surveillance. B7-H3 acts on two sides, inhibiting the T and NK cell-mediated antitumor activity by reacting with a still unknown receptor, and favoring tumor progression by promoting migration, invasiveness, and drug resistance [52,59,60]. For these reasons, B7-H3 represents a consolidated negative prognostic marker in several adult and pediatric tumors including neuroblastoma (NB) [61]. In particular, in primary NB, high B7-H3 surface expression also correlates with poor survival in patients with localized disease, indicating that the analysis of its expression could improve patients’ risk stratification [60,61].

Different therapeutic strategies targeting B7-H3 have been explored in preclinical studies with promising results [49,62,63,64,65,66]. Phase I clinical trials based on the infusion of humanized anti-B7-H3 monoclonal antibodies (mAbs) have been completed in adult and pediatric tumors including NB (NCT02982941), with results supporting the design of phase II and III clinical trials.

Whereas all the previously described molecules mainly impair lymphocyte-mediated immune surveillance, CD47 and its ligands, thrombospondin-1 and signal regulatory protein α (SIRPα), represent an inhibitory axis limiting phagocyte activity. Different from B7-H3, which can be considered a tumor-associated antigen, CD47 is overexpressed by many types of tumors but is also widely expressed in normal cells. The interaction of CD47 with SIRPα gives macrophages a “don’t eat me” signal, inhibiting phagocytosis and allowing tumor cells to evade immune surveillance. The CD47/SIRPα axis is emerging as a key immune checkpoint in different cancers including hematological malignancies. This drives the development of immunotherapeutic strategies aimed to disrupt this brake. Importantly, however, due to the broad expression of CD47 on healthy cells, deep preclinical and clinical studies proving the safety of this therapeutic approach are required.

### 2.2. Soluble Mediators and Therapeutic Approaches

#### 2.2.1. TGF-β and IL-10

Several soluble mediators are establishing an immunosuppressive milieu within the TME. Cytokines, growth factors, and metabolites, eventually packed into extracellular vesicles such as exosomes, play a central role in the intricate networking between cancer and immune cells, as well as between the different immune cell subsets.

Suppressive cytokines are either produced by tumor cells or immune cells having an immunosuppressive/pro-tumoral activity such as regulatory T cells (Tregs), myeloid-derived suppressor cells (MDSCs), and type 2 polarized tumor-associated neutrophils (N2, TANs) or macrophages (M2, TAMs). TAMs heavily contribute to tumor progression, exerting a suppressive and opposite role as compared to their proinflammatory M1 counterpart [67]. Among the cytokines involved in the generation of a suppressive microenvironment, TGF-β and IL-10 are known to play a central role. TGF-β has a direct pro-tumor effect on cancer cells and promotes the exhaustion of immune responses in different types of cancer [68]. In particular, TGF-β suppresses NK cells through multiple mechanisms. These include the direct inhibition of the mTOR pathway, impairing NK cell activation and function [69,70], the downregulation of the expression of different activating receptors including NKp30, NKG2D [71], DNAM-1, and CD16 [72,73], and the modulation of ligands on target cells [74,75]. TGF-β also modifies the chemokine receptor repertoire of NK cells, likely impacting their recruitment at the tumor site [76,77], and promotes the generation of NK cells with a low cytotoxic ILC1-like phenotype [78]. Interestingly, unlike other typical immunostimulatory cytokines such as IL2, IL-12, and IL-15, IL-18 potentiates rather than suppresses some of the TGF-β-mediated modulatory effects [79]. Besides the classical soluble form, recent findings show that the antitumor function of NK cells can also be suppressed via the contact with membrane-bound TGF-β expressed on metastasis-associated macrophages or Tregs [80,81].

Regarding the regulatory properties of TGF-β on T cells, the cytokine has been demonstrated to inhibit the differentiation of T cells toward the antitumor Th1 phenotype, inhibit their proliferation through IL-2 downregulation, and impair the cytotoxic effect of CD8+ T cells through the repression of granzyme B and IFN-γ. Moreover, TGF-β is involved in the upregulation of FoxP3 in CD4+ naïve T cells, inducing their differentiation toward Tregs [82,83,84,85], and, accordingly, TGF-β blockade results in Treg depletion in different cancers [86,87]. Importantly, TGF-β has also been correlated with resistance to the immune checkpoint blockade, as demonstrated by Hugo et al. through transcriptomic analysis on metastatic melanoma specimens [88]. Thus, TGF-β blockade can also be considered as a strategy to enhance the efficacy of therapies including the inhibition of immune checkpoints [89,90] and the adoptive transfer of CAR-engineered T cells [91,92]. An attractive approach is the combined targeting of immune checkpoint molecules and TGF-β within the same moiety, which has been demonstrated to be more effective in vivo than the single targeting [93]. Along this line, in a recent paper, Chen et al. engineered CAR-T cells secreting a bispecific trap protein binding PD-1 and TGF-β, demonstrating a significant improvement in effector T cell engagement, persistence, and expansion, preserving CAR-T cells from exhaustion, and leading to high antitumor efficacy and long-term remission in animal models [94].

IL-10 is a cytokine, mainly produced by Tregs, B cells, dendritic cells (DCs), and macrophages, suppressing the function of antigen-presenting cells (APCs) and CD4+ T cells [95]. The immunosuppressive role of IL-10 has been attributed to the downregulation of IFN-γ, the impairment of DC maturation, and the downregulation of HLA-I, on cancer cells, and HLA-II and costimulatory molecules (CD80 and CD86), on APCs [96,97,98]. Moreover, Ma et al. recently demonstrated that the over-production of IL-10 converts lymphoma-associated Th1 cells into FoxP3-negative/PD-1-overexpressing T regulatory type 1 cells, generating an immune escape signature [99]. IL-10 can also induce a pro-tumor phenotype in macrophages during the early phases of tumor formation, as demonstrated by Michielon et al. in 3D organotypic melanoma cultures [100]. Similar to TGF-β, IL-10 expression could be considered as a predictive biomarker of response to the blockade of immune checkpoints, especially when considering the IFN-γ/IL-10 ratio.

#### 2.2.2. PGE2 and Metabolites

Specific classes of prostaglandins (PGs), molecules involved in inflammatory processes, have been associated with cancer development and progression. Namely, PGE2 contributes to the immunosuppressive tumor milieu. For example, produced by melanoma-associated fibroblasts, PGE2 negatively regulates the expression of NKp44 and NKp30 activating receptors in NK cells [101]. In breast cancer, it has been found to be associated with reduced CD80 expression on macrophages, thus hindering the antitumor immune response, and the administration of ibuprofen in vivo led to tumor shrinkage, active recruitment of T cells, and the reduction in immature monocytes [102,103]. Recently, the COX2/PGE2 pathway has been associated with M2 polarization of macrophages in hepatocellular carcinoma patients, and M2 macrophages were found to inhibit the production of IFN-γ and granzyme B from CD8+ T cells both in vitro and in vivo [104]. Therefore, the administration of PGE2-inhibiting drugs might help in the re-education of the TME. Studies performed on syngeneic mouse models revealed that the combinatory administration of a PGE2 receptor antagonist and PD1 blockade had a synergistic effect, leading to a massive reorganization of the tumor immune environment [105].

Studies in skin squamous cell carcinoma reported that PGE2 is associated with increased tumor cell migration and invasion, correlating with the staging [106]. This correlation has also been reported in gliomas, where PGE2 seems to be involved in the promotion of the tryptophan-2,3-dioxygenase pathway, known to mediate tolerogenic signaling through multiple mechanisms [107,108].

The accumulation of kynurenine due to tryptophan catabolism leads to its binding to AhR, further exerting an immunosuppressive pressure. The AhR nuclear translocation results in the upregulation of FoxP3 and IL-10 in T cells, driving the acquisition of a regulatory phenotype, reducing the immunogenic capacity of DCs [109,110,111,112], and upregulating PD-1 expression on effector T cells [113]. Importantly, the inhibitory effect of kynurenine has also been well documented in human NK cells [114]. In particular, l-kynurenine hampers the cytokine-mediated strengthening of the NK-cell-mediated killing, limiting the upregulation of NKp46 and NKG2D receptors. As a consequence, NK cells conditioned by l-kynurenine display a reduced ability to kill target cells mainly recognized via these receptors. Given all these observations, it is not surprising that different therapeutic approaches targeting the Trp-Kyn-AhR pathway are currently in preclinical development or clinical trials, in combination with standard therapies [115].

Another important metabolite exerting an immunosuppressive effect is adenosine, which can be generated within the TME due to the over-secretion by tumor cells of ATP and its catabolism by specific ectoenzymes. Intracellular ATP, produced by glycolytic or oxidative metabolism, can be released in the extracellular space through passive efflux or active secretion [116]. A “passive” release, due to a high intracellular concentration, can be associated with cytotoxicity, meaning that ATP represents a cell damage marker. The active secretion occurs through exocytosis or membrane transporters such as the ABC (ATP-binding cassette) proteins and is triggered by events such as hypoxia [117]. Importantly, hypoxia also induces the overexpression of CD39 and CD73 ectoenzymes, promoting the conversion of ATP to AMP and AMP to adenosine, respectively, as well as the downregulation of adenosine kinase, limiting the conversion of adenosine in its final metabolites and leading to the accumulation of adenosine in the extracellular space [118]. The ectonucleotidases can be expressed by tumor cells and different subsets of innate or adaptive immune cells [116,119,120]. Moreover, it has been reported that tumor cells and tumor-derived exosomes can carry CD39 and CD73 on their membranes, thus promoting ATP conversion and adenosine accumulation in the TME [120,121].

Adenosine also promotes the conversion of macrophages toward the immunosuppressive M2 phenotype, and the release of MMPs by tumor-associated neutrophils, thus favoring the invasive and metastatic process [122,123,124,125]. In Tregs, the activation of A2A receptors induces the proliferation, activation, and overexpression of the CTLA-4 and PD-1 immune checkpoints [126].

Finally, another molecule that is significantly involved in the generation of a pro-tumor setting is the vascular endothelial growth factor (VEGF), highly secreted by tumor cells and by pro-tumoral TAMs [127,128,129,130].

## 3. Three-Dimensional Culture Models: Moving from Spheroids to Next-Generation 3D Tools

Despite the significant information described in the previous sections and obtained by using 2D cultures and animal models, few tumor escape mechanisms have been addressed in 3D platforms, pointing out the need to move quickly towards these more reliable 3D culture systems.

These systems represent next-generation 3D tools that are slowly replacing spheroid-based strategies widely employed thus far to investigate tumor-immune system interactions in vitro and that still represent one of the gold standard 3D models to assess tumor–immune cell interactions. In particular, these tools have allowed us to gain insights into T and NK cell-related infiltration, cytotoxicity, and soluble factor release [131,132,133,134,135,136,137,138]. Moreover, spheroids have allowed deepening the understanding of the effects of the TME on macrophages’ polarization and functions [139,140,141].

Despite these encouraging outcomes, the absence of an ECM limited the reliability of such spheroid-based systems and hampered the evaluation of the effects of chemical–physical properties of the surrounding microenvironment on cell activity [142,143]. Therefore, many researchers and material scientists are moving towards scaffold-based 3D platforms, which can also be integrated with different components of the TME. In the following sections, we will recapitulate the state of the art of 3D hydrogel-based in vitro models that have been adopted to study tumor and immune system interactions as well as novel immunotherapeutic approaches (summarized in Table 1), describing the different types of biomaterials that have been employed (schematically reported in Figure 2a). Then, we will introduce recent emerging immune-on-chips that are paving the way for the assessment of more predictive 3D models.

### 3.1. Natural Biomaterial Tools for 3D Tumor Modeling In Vitro

Biomaterials of natural origin are the most employed materials in several biomedical applications, due to their high biocompatibility, bioactivity, mechanical and biochemical properties similar to those of the ECM in vivo, and the presence of chemical cues promoting cell attachment and proliferation, reciprocal communication, and tumorigenesis, thus being a gold standard in cancer research [144,145]. They can be assigned to two main categories: (i) protein polymers, and (ii) polysaccharide polymers.

#### 3.1.1. Protein-Based Polymers

Among biomaterials for 3D tumor modeling in vitro, the most adopted is Matrigel, which is an extract of the basement membrane matrix of Engelbreth Holm Swarm mouse sarcoma. This commercially available ECM, which generates a hydrogel at 24–37 °C, has a very similar content to the in vivo counterpart as it comprises various ECM macromolecules such as collagen IV, fibronectin, laminin, and proteoglycans, as well as different growth factors, chemokines, cytokines, and proteases [146,147]. Due to these constituents, Matrigel represents a biologically active platform able to promote the adhesion, migration, and differentiation of different cell types in vitro. Therefore, being easily available and versatile, and applicable with a wide variety of cellular phenotypes, it represents a standard support matrix for cell culture in several biomedical applications. In particular, it has been largely employed in 3D tumor modeling for the investigation of cancer progression, angiogenesis, metastasis, and drug efficacy [146]. Tumor cells are extremely proliferative in Matrigel-assisted cultures, differently from normal cells, showing an in vivo-like invasive profile. It has also been proved that 3D Matrigel matrices allow cells to express fundamental features related to their intrinsic malignancy [143]. For example, in the case of breast cancer, it is possible to discern poorly or highly aggressive cells when they are encapsulated within this hydrogel by examining their morphology. They usually organize into small aggregates (i.e., acini-like structures, luminal phenotype) or display an elongated shape with pronounced extensions (basal phenotype). Malignant cells are also capable of migrating through Matrigel matrices by enzymatic degradation, which is commonly studied through the Boyden chamber assay [143].

It is widely recognized that the cancer invasive profile also correlates with the ability of tumor cells to evade the immune surveillance as well as driving different types of immune cells to participate in cancer progression through cell–cell contacts and release of soluble factors [148,149]. For example, Ramirez et al. demonstrated that malignant cancer cells are capable of inducing macrophages to change the gene expression profile. Indeed, in a 3D Matrigel-based system, the interplay between the human macrophage U937 cell line and breast tumor cells caused, in U937, a significant upregulation of MMP1 and MMP9, both involved in tumor invasion via ECM degradation. Moreover, an upregulation of the inflammatory COX2 gene inducing the pro-tumoral factor PGE2 was observed. Such increments were significantly higher in the co-cultures of U937 with MDA-MB-231 cells, a highly aggressive triple-negative breast cancer cell line, than with MCF7, which has characteristics of a differentiated mammary epithelium [150]. The same group, in a later work, showed that primary breast cancer cells constitutively secrete high levels of CCL5, CCL2, and G-CSF, specifically involved in the attraction of circulating immune cells at the tumor site, while a remarkable increase in IL-1β, IL-8, MMP-1, MMP-2, and MMP-10 production was revealed when cancer cells were co-cultured with monocytes [151].

Taken together, these data support the idea that tumor aggressiveness is related to its capability to shape the inflammatory microenvironment by recruiting immune cell populations at the tumor site and instructing them to fulfill pro-tumoral functions. Therefore, it is evident that the interactions occurring within the TME between different cell types, including stromal cells, play a fundamental role in promoting disease progression [152,153].

Hence, tissues explanted during surgical resections or biopsies have been embedded in Matrigel to investigate immune cell populations infiltrating the tumors [154,155]. For instance, in slices derived from tissues of patients with colorectal and lung cancer, a great presence of myeloid-derived suppressor cells (MDSCs) (CD206+/CD33+/HLA-DR–-) and CD4-/CD8-T cells, as well as a reduced number of NK cells and monocytes, has been observed [156]. Moreover, innovative organotypic cultures have been adopted by co-culturing organoids established from patient-derived cancer cells (due to their capability of retaining key pathophysiological and structural features of the original tumor in vitro [157]), with patient-matched stromal (e.g., CAFs) and immune components (e.g., T cells) [158]. These systems represent a valuable tool for studying the complex tumor–stroma–immune system communications in a highly reliable context, paving the way for the assessment of novel personalized immunotherapeutic strategies. To this end, more recently, Dijkstra et al. co-cultured autologous colorectal or non-small lung cancer tumor organoids with peripheral blood lymphocytes, with the intention of increasing the number of tumor-specific CD8+ T cells to be infused in patients [154]. Furthermore, other groups focused on testing novel engineered immune cell-mediated strategies. Among them, αβT cells modified to express a tumor-specific γδ TCR (TEGs) were used in primary myeloma cells grown within a 3D BM niche model [159]. Moreover, the CAR-NK-92 cell line was proposed as an effector against patient-derived colorectal cancer organoids by targeting the epidermal growth factor receptor variant III (EGFRvIII) [160], overexpressed in a wide variety of epithelial tumors [161]. Moreover, researchers are adopting such patient-derived preclinical platforms to evaluate different strategies targeting immune checkpoint axes, alone or in combination. In this latter context, an association of an anti-PD-L1 mAb (atezolizumab) with MEK inhibitors (selumetinib) led to a higher MHC-I expression on non-small lung cancer organoids, together with increased secretion of IFN-γ, IL-6, IL-1β, and TNF-α by immune cells [162].

However, despite all the encouraging results derived from in vitro and in vivo preclinical models, many patients do not respond to some promising therapies, even due to the great variety of mechanisms involved in cancer immune evasion that are still not completely understood. Furthermore, although Matrigel establishes a favorable TME [163], it is affected by several drawbacks that considerably limit its use. Firstly, because of its structural weakness, it is mainly adopted as a monolayer or a thin gel conformation, principally for short-term invasion assays [143]. Then, the applicability of Matrigel is severely hampered because of its variability in composition and structure, due to its natural origin (e.g., tumor sizes from which is extracted, prepared, etc.) [147]. Differences in mechanical and biochemical properties between the various batches and within a single batch negatively impact the experimental reproducibility [147,164]. These constraints, along with the fact that Matrigel is difficult to manipulate physically and biochemically, make comparisons between and within laboratories remarkably challenging [164,165]. Moreover, being an animal-derived ECM, the presence of xenogenic contaminants may hamper the use of Matrigel-based cell cultures as in vitro preclinical tool for screening effective immunotherapies. For instance, lactate dehydrogenase elevating virus (LDHV), a mouse virus capable of infecting macrophage cells, possibly influencing both the immune system and tumor behavior, was detected in multiple batches of Matrigel [166].

All these considerations should be kept in mind when interpreting results based on Matrigel-assisted cell cultures, to distinguish biological effects caused by controlled experimental conditions or variables from the hydrogel itself [164].

Collagen is another biomaterial belonging to this category that is largely employed as an ECM-supporting matrix for 3D models, as it contains fundamental cellular adhesion domains (i.e., arginine-glycine-aspartate (RGD) peptide) that favor cell growth in vitro. It is commonly deposited by different cancer types during malignant progression, thus being an important component of the TME. In particular, matrices made of collagen type I promote, in vitro, uncontrolled cancer cell growth, the establishment of hypoxic regions, and angiogenesis, thus being particularly suitable to resemble key environmental properties of tumors [167,168]. Considering this, several studies have been conducted to reproduce the complexity of the TME by including, in 3D collagen constructs, cancer cells with components of the tumor stroma as well as immune cells in close contact with each other. Cell-to-cell contact is notably critical when evaluating the anticancer activity of cytotoxic lymphocytes, which requires direct interactions with tumors to efficiently kill malignant cells [169]. Moreover, as discussed before, an immune-mediated pro-tumoral action is frequently observed within the TME, particularly due to the presence of TAMs supporting cancer progression and resistance to chemotherapies. For example, macrophages co-cultured with breast cancer cells in a more in vivo-like environment led to a significant increase in oxygen consumption as well as in the secretion of epidermal growth factor (EGF) and IL-10, suggesting a synergistic crosstalk between different types of cells and indicating a tumor-promoting activity of immune cells colonizing tumors such as M2-polarized macrophages [170]. It was demonstrated that macrophages’ polarization towards an M2 phenotype is reached spontaneously in organotypic co-cultures including cancer cells and fibroblasts after three weeks, with a consequent reinforced proteolytic activity of the tumor cells through the increase in MMP2 and MMP9 production. Moreover, the same authors showed that organotypic co-cultures allow handling either M1 or M2 polarization via stimulation with IFN-γ and LPS or IL-4, respectively [171]. This can help to deeply elucidate the role of macrophages in the TME, where they can contemporarily show a tumor-promoting effect or exert an antitumor activity by attacking and eliminating cancer cells, depending on their polarized status [172]. Therefore, the importance of developing more reliable in vitro systems taking into account the complex reciprocal interactions occurring in vivo between malignant and non-malignant cells is evident. Recently, some platforms prepared the groundwork for the investigation of novel agents (e.g., immunotherapeutic antibodies) aimed at targeting the key cellular components of the TME (e.g., CAFs or TAMs) in a clinically relevant context [173].

Overall, collagen has been widely employed as an EMC-mimicking matrix in the field of cancer research, also due to its easy manipulation and low costs, making this biopolymer easily accessible to the scientific community [174]. Despite its intrinsic poor mechanical properties, it can be easily tuned by changing the concentration or adding synthetic crosslinking agents to finely tune its structure and stiffness based on the specific application [167,168,175]. However, because of its animal origin, as with Matrigel, it is affected by risks associated with biological materials, such as the batch-to-batch variability, that limit the reproducibility of the results [174].

#### 3.1.2. Polysaccharide-Based Polymers

Polysaccharide-based biopolymers have been largely adopted as ECM-supporting matrices for in vitro cell culture since they are characterized by low immunogenicity as well as elevated biocompatibility [165]. Several biomaterials belonging to this group have been used to support cancer cells’ interactions with the immune system, especially focusing on those mechanisms occurring within the TME that promote tumor growth and metastasis [176]. Among these polymers, alginate is one of the most employed. Alginate, derived from brown seaweeds, presents a molecular structure comparable to that of polysaccharides found in vivo [177]. It is particularly suitable for the formation of cell-laden microspheres, allowing for obtaining a high number of replicates due to its easy manipulation, fast gelation, thermal stability, and low cost [165,167,177,178].

In our recent publication, we selected alginate for developing a 3D model of NB. Both MYCN- and non-MYCN-amplified cell lines showed cellular proliferation, drug sensitivity, and immunophenotypic plasticity closer to those occurring in vivo, with respect to 2D models. Intriguingly, we observed molecular characteristics more similar to immunophenotypic variances occurring in vivo and not fully comprehended in traditional 2D culture conditions, such as the IFN-γ-induced negative regulation of PVR (CD155) expression on tumor cells after 7 days of 3D culture [23]. Moreover, it was possible to appreciate an IFN-γ-dependent upregulation of the immune checkpoint ligand B7-H3, a molecule deeply discussed above [179].

Alginate microencapsulation has also been used to explore the onset of either a proinflammatory or an immunosuppressive TME, especially focusing on the dynamic interactions occurring between the main cellular components that support the tumor malignant behavior [180]. In a 3D co-culture of non-small cell lung carcinoma cells with CAFs and monocytes, an accumulation of soluble factors (IL-4, IL-10, IL-13, CXCL1) was observed, promoting immune cell infiltration of the tumor and M2-like macrophage polarization. This polarization was characterized by the expression of the CD68, CD163, and CD206 markers and the production of the CCL22 and CCL24 chemokines [181].

Chitosan is a linear polysaccharide derived from the partial deacetylation of chitin, which is abundantly available from different biological sources, being, for example, the main structural polymer of crustacean exoskeletons [177]. Due to its poor solubility in common solvents, the process of extraction of chitin is quite laborious, thus limiting its utilization. In general, chitosan offers a higher mechanical strength, and the possibility to be easily chemically modified, and to interact with other biomolecules due to the presence of reactive functional groups. Furthermore, it simply forms soft gels and crosslinks with other polymers [182,183]. Besides these characteristics, chitosan represents an effective alternative candidate for 3D cultures of cancer cells due to a structure similar to that of GAGs, one of the main constituents of the tumor ECM [177,184]. The chitosan and alginate (CA) combination has also been largely adopted to realize porous scaffolds that exhibit better mechanical strength and shape maintenance when compared to chitosan alone, because of the electrostatic contact between chitosan’s amine groups and alginate’s carboxyl groups [177,184]. Three-dimensional CA scaffolds provide a cost-effective feasible model to evaluate in vitro the interplays between tumors and the immune system in a clinically relevant context [185]. For example, these platforms can mimic the breast cancer TME. In this context, the inactivation of CAFs, which have been demonstrated to induce T cell suppression in breast tumor stroma [186], or combined gene therapies aimed at enhancing T cell infiltration and activation in the tumor milieu [187] may represent novel strategies for improving the efficacy of the current adoptive T cell therapies against breast cancer.

However, there are also different drawbacks associated with these types of biomaterials. For example, chitosan is characterized by poor mechanical properties [178], and alginate by a variable degradation rate. Moreover, the latter does not possess integrin-binding sites, thus often requiring chemical modification or conjugation with other bioactive polymers [167]. Indeed, extensive literature has been reported on the covalent functionalization of alginate with the RGDpeptide to favor cellular adhesion, proliferation, and migration [167,168,188,189].

In conclusion, natural polymers are highly suitable to recapitulate in vitro the main features of the native ECM. Nevertheless, they suffer from important limitations. Besides the aforementioned significant batch-to-batch variability (e.g., various mechanical and biochemical features, peptide or protein concentrations) and xenogeneic contaminations associated with polymers derived from an animal source, it is generally difficult to control scaffold degradation rates, possibly influencing cellular activity in unknown ways [144]. Moreover, natural polymers can be realized in a limited range of mechanical stiffness, porosity, or biochemical cues [145].

Therefore, the focus is shifting toward synthetic polymers that may mimic the biomimetic qualities of natural ones while providing more repeatability and control over the materials’ physical and chemical properties.

### 3.2. Synthetic Biomaterial-Based 3D Tools

Synthetic polymer-based scaffolds represent a valid alternative to naturally derived ones. First, being free of xenogeneic and possible contaminants, they enable high reproducibility by reducing inter-batch variations, thus resulting in a greater consistency of the results. Moreover, they can be more easily manipulated to finely tune mechanical and chemical properties as well as degradation rates for specific cell culture applications [144,164,178,190]. Indeed, even though synthetic biomaterials are biologically inert, allowing, but not promoting, cellular activity, cell adhesion ligands and other bioactive molecules can be precisely introduced via covalent attachment, adsorption, or electrostatic interactions depending on the desirable environmental cues that need to be investigated [145,178,190,191]. Therefore, emerging studies have demonstrated the possibility to adopt these polymers for investigating the interactions between cancer cells and the immune system within the TME in vitro. For instance, 3D polystyrene-based scaffolds have been used to mimic T cell infiltration in non-small lung cancer and to explore the subset of inflammation proteins related to the co-cultivation of tumor cells with lymphocytes [192], while polycaprolactone (PCL) has been exploited for examining the capability of DCs to engulf dying colon cancer cells through the same mechanisms observed in the human body [193].

One of the most common synthetic polymers is polyethylene glycol (PEG), which has been largely used in the tissue engineering field both in vitro and in vivo, showing to be highly suitable as a model for ECM–cancer interaction studies [145]. It is biocompatible and fully hydrated, thus closely reproducing the soft tissues’ characteristics, and particularly suitable for cell encapsulation due to the liquid-to-solid transition to form hydrogels encapsulating cells [184]. Even though it is biologically inert, it can be easily functionalized with protease-sensitive peptides to render the surrounding ECM enzymatically degradable by cells. Numerous studies have explored the inclusion, via crosslinking reactions, of different peptides sensitive to MMP-mediated cleavage, in order to evaluate cell migration and invasion [194,195]. The presence of MMP-cleavable sites in PEG hydrogels has also been shown to promote cell proliferation and differentiation [196,197,198,199]. Moreover, cellular adhesion and/or other molecules of interest (VEGF, TGF-β1, etc.) can been integrated through various non-toxic polymerization techniques [164,184,200]. Interestingly, in a recent study, the migration and function of NK-92 cells within a 3D RGD-functionalized PEG hydrogel containing either non-small lung cancer metastatic (H1299) or non-metastatic (A549) cell lines were investigated. The metastatic tumor model displayed a greater loss of stress ligands (ULBP1, MICA), downregulation of chemokine expression (MCP-1), and higher production of inhibitory soluble molecules (i.e., TGF-β, IL-6), as compared with a non-metastatic tumor model, more resembling the in vivo scenario. The NK cell migration toward cancer cells and their co-localization depended on the immunomodulatory profile of tumors, and NK-92 cells decreased the production of RANTES and MIP-1 α/β when incubated with H1299 cells. The study highlights the benefits of 3D cancer models that allow us to examine the effects of signals on NK cell migration. In addition to the release of soluble substances, immune cell infiltration might be influenced by the physical features of tumors. Nevertheless, the impact of matrix stiffness on NK cell migration is unknown, and more research is needed to fully understand the NK cell mechanotransduction pathways [201].

Despite recent promising outcomes, PEG and other synthetic biomaterials are still poorly adopted in cancer research. Despite the fact the raw materials for making PEG hydrogels are about half the price of Matrigel, the necessity for one or more synthetic peptides to provide the essential biochemical cues to drive cellular behavior can be prohibitively expensive for large-scale manufacturing. Furthermore, extensive adjustments to obtain the desired combination of physical and biochemical properties driving cellular behavior can be time-consuming, costly, and challenging [164], whereas degradation products are often non-biocompatible [202]. Finally, when compared to in vivo tumors, cells cultivated in completely synthetic platforms can proliferate without some tumor-like gene expression patterns, revealing inconsistent tumorigenicity and metastatic potential, or resistance to anticancer treatments. As one might expect, such difficulties have an impact on the creation of reliable tumor-mimicking 3D in vitro models [167].

To overcome these disadvantages and achieve more in vivo-like conditions, synthetic materials can be properly mixed with naturally generated biopolymers [202], in order to better address the physiological crosstalk between immune and cancer cells.

**Table 1 cancers-14-01013-t001:** Summary of 3D in vitro models based on different types of polymeric matrices to study cancer–immune interactions and immunotherapies.

3D Biomaterial	Cell Types	Main Objectives	Ref
Matrigel	Breast cancer cells, NK and Treg cells	To compare tumor biomarkers’ expression and immune infiltration between luminal and basal tumor phenotypes	[149]
	Breast cancer cells, promonocytic cells/monocytes	To study tumor/immune cells’ crosstalk	[150,151]
	Colorectal and non-small lung cancer spheroids, peripheral blood lymphocytes (PBLs)	To obtain patient-specific tumor-reactive T cells	[154]
	Colorectal or lung cancer tissues	To maintain primary cells in culture and study tumor-infiltrating immune cell populations	[156]
	Pancreatic cancer organoids, CAFs, PBLs	To analyze multiple cells’ crosstalk	[158]
	Endothelial progenitor cells, multipotent mesenchymal stromal cells, CD138+ myeloma cells	To study engineered (to express a defined γδTCR) T cells’ activity	[159]
	Colorectal cancer organoids	To study CAR-NK cells’ activity	[160]
	Non-small lung cancer cells, peripheral blood mononuclear cells (PBMCs)	To establish an effective combined therapy based on MEK inhibitors and anti-PD-L1	[162]
Collagen	Pancreatic tumor spheroids, T cells	To monitor cancer invasive behavior and T cell cytotoxicity	[169]
	Breast cancer spheroids, macrophages	To investigate macrophages’ polarization, localization, and function in the tumor mass	[170]
	Squamous carcinoma cells, fibroblasts, macrophages	TME-mediated regulation of macrophage polarization, both spontaneous and induced by exogenous factors	[171]
	Lung adenocarcinoma cells, fibroblasts, macrophages	To analyze multiple cells’ crosstalk	[172]
	B lymphoma cells, fibroblasts, macrophages	To reproduce the lymphoma microenvironment to test therapeutic Abs	[173]
Agarose	Hepatocellular carcinoma cells, M2 macrophages	To investigate the impact of macrophages on cancer progression	[176]
Alginate	MYCN- and non-MYCN-amplified NB cells	To analyze tumor immunophenotype related to NK cell receptors	[23]
	Breast cancer cells, fibroblasts, and macrophages	To analyze multiple cells’ crosstalk	[180,181]
	Non-small cell lung carcinoma cells, CAFs and monocytes		
Alginate/ Chitosan	Prostate cancer cells, PBLs	To study tumor/immune cells’ crosstalk	[185]
	Mammary carcinoma cells, CAFs, T cells	To evaluate the impact of CAFs on T cell function	[186]
	Mammary carcinoma cells, T cells	To explore how tumor CCL21 and IFN-γ expression affects T cell recruitment and activation	[187]
Polystyrene	Non-small lung cancer cells, T cells	To study tumor/immune cells’ crosstalk	[192]
PCL	Colon cancer cells, DCs	To study tumor/immune cells’ crosstalk	[193]
PEG	Non-small lung cancer cells, NK-92	To study NK cells’ infiltration and function	[201]
PEG/Chitosan	Mammary carcinoma cells, T cells	To study the influence of TME on drug efficacy and immune resistance	[202]

## 4. Micro-Physiological Systems for Investigating Immune Cell–Tumor Dynamic Interactions

Cancer immunotherapy has shown many signs of progress in recent years, thanks to novel strategies aimed at enhancing the efficacy of immunomodulating agents and their patient-specific approaches. Furthermore, alternative in vitro platforms are in continuous development for overcoming some limitations of traditional preclinical models. As we discussed above, 3D tissue models allow us to better investigate tumor pathways in a tissue-like architecture than in simplistic cell culture monolayers, by introducing physiological barriers that mediate immune system–cancer cell crosstalk. However, the three-dimensionality itself cannot properly reproduce the complexity of the human tissues and organs, where dynamic stimuli (e.g., blood and lymphatic flow mechanical forces) shape the immune cell infiltration and their dynamic interactions with cancer cells. In this context, recent advancements in the realization of immunological tissues-on-chips have increased the relevance of in vitro models, leading to better knowledge of immune cell recruitment, selection, invasion, and activation within the tumor milieu [34,203].

Microfluidics is a rapidly growing technology using narrow channels, ranging in height/width from tens to hundreds of micrometers, to study cell migration by handling small fluid volumes, thus reducing the amounts of reagents and biological materials. Overall, microfluidic assays outperform other in vitro models in terms of physiological relevance because they allow local monitoring of the cellular, physical, and biochemical cues, making them a good compromise between in vivo and other types of in vitro systems [204,205]. They may also permit real-time imaging with fine control on the interplays between different cellular populations growing within the interconnected channels. This enables the investigation of spatiotemporal dynamics similar to those found in the TME, as well as the effects of environmental characteristics on cell behavior such as changes in pH and oxygen levels (i.e., acidification and tumor hypoxia) and cytokine/chemokine gradients [206]. Several studies focused on cancer and immune cell interactions as well as on cell-mediated immunotherapies used microfluidic devices coupled with 3D hydrogel-based models. The most recent and relevant publications are reported and summarized in Table 2.

Despite the benefits discussed above, some technical and biological limitations still affect these platforms. First, most of them are fabricated with polydimethylsiloxane (PDMS) that may cause toxicity due to the progressive release of non-crosslinked oligomers, and the retention of small hydrophobic molecules through adsorption, making some biochemical analyses difficult [204,206,208,212,213,215,220]. Furthermore, although the recovery of cells from the separate compartments of microfluidic devices is feasible, the low number of recovered cells makes it difficult to perform functional and phenotypic analysis [204]. For this reason, most data are derived from imaging analysis. Furthermore, in the case of immune cells loaded in microfluidic tools, monitoring the functional and phenotypic changes occurring over time without interrupting and altering the microfluidic flux is not feasible.

Moreover, the clinically relevant size of the tissue models hosted and cultured in such microfluidic devices still represents a challenge. Extended research in properly resembling the ECM complexity within these devices for the establishment of a tumor niche and the co-culture of different immune cell types (MDSCs, Th1 cells, Bregs, eosinophils, etc.) should be carried out [34,204].

Interestingly, at this time, the majority of these microfluidic devices are adopted for co-culturing tumor–immune cells in different compartments, physically separated by a polymeric gel, for evaluating the real-time cell migration under virtual static conditions, similar to the Boyden chamber-based assays [204,206,208,212,213,215,220]. However, these conditions do not properly recapitulate the circulatory flow in the human body, limiting the experience of blood flow-associated forces (e.g., shear stress) that influence the survival, escape, and activation of immune cells during their journey in the vascular network [142,224]. Along this line, the authors of this review are currently involved in the study of the NK-mediated cytotoxicity against NB cell-laden hydrogels [23] within the novel fluidic device MIVO^®^ (Multi in Vitro Organ System), which is schematically represented in Figure 2b, capable of both mimicking the blood flow circulation in a highly reliable context [225,226,227], and culturing clinically relevant sized cancer tissues, as previously reported in recent works [226,227].

## 5. Conclusions and Future Directions

The recent assessment of innovative immunotherapies against cancer led to the urgent need for an increasing number of predictive preclinical models capable of reproducing the key features of the TME and properly testing the efficacy of immunomodulating agents. Up to now, the gold standard still relies on in vitro cell culture monolayers and animal testing, which, over the years, enabled us to take important steps forwards to the knowledge of tumor immunology. However, it is increasingly evident that novel platforms should be realized for overcoming the limitations that affect the traditional settings, which unfortunately often result in ruinous discrepancies between the benefits observed in preclinical and clinical studies.

In the present review, we have provided an overview of 3D in vitro models based on different types of biomaterials more commonly used to study tumor–immune cell interactions, and standard and innovative immunotherapeutic strategies. As discussed, the composition of the TME can play a pivotal role in malignant progression due to the evasion of the immune surveillance. Depending on their own characteristics, which may be more or less suitable for some specific applications, both natural and synthetic polymeric matrices are currently adopted to reproduce the key features of the tumor-surrounding ECM.

In general, typical ECM organic components, such as GAGs, proteoglycans, and glycoproteins interact with cancer, stromal, and immune cells and highly impact the hydration and stiffness of the ECM. Thus, to provide cells with such biological signals, bioactive polymers such as Matrigel or chitosan can be selected, while inert polymers such as alginate or PEG need to be functionalized to become biologically active. Other ECM elements, such fibrillary components, greatly impact tumor progression, also representing physical barriers both for tumor and immune cells. Thus, a polymer enriched in fiber elements may represent a good tool to investigate cancer cell invasiveness and metastasis as well as the mechanisms regulating immune cell recruitment in the tumor mass. It should be taken into consideration that the ECM can be shaped by the action of radio- and chemotherapy. In this context, in a recent study focused on lung cancer, chemotherapy-induced remodeling of the ECM occurred with the pivotal intervention of host T cells which, stimulated by paclitaxel, increased their release of lysyl oxidase, favoring the formation of tumor metastases [228].

It is important to highlight that the ECM composition and stiffness greatly vary among different types of tumors. This should be considered when setting an appropriate 3D model. Indeed, some solid tumors are characterized by a more rigid ECM, thus requiring hydrogels with high stiffness. In this case, an optimal choice may be represented by synthetic polymers that allow finely tuning their composition to reach a proper stiffness. Conversely, other materials such as collagen or chitosan form hydrogels characterized by poor mechanical properties. On the other hand, other malignancies arising, for example, in soft tissues may be modeled with low-stiffness and highly hydrated polymers such as alginate. Therefore, the choice of the most appropriate biomaterial-based model strictly depends on the type of tumor to be addressed and on the experimental aims.

Despite encouraging results, we are still far from the possibility to explore cancer immunotherapeutic approaches in a system that closely resembles the complexity and dynamic of the human body. Microfluidic tools represent a challenging approach in this sense, by introducing dynamical cues and the possibility to culture cells in a highly controllable environment that may help to elucidate the mechanisms exploited by immune cells to infiltrate tumors.

Some of the mentioned issues such as the experimental limits of microfluidic devices can be overcome by designing new fluidical platforms that can be adapted to the standard laboratories’ tools commonly used for in vitro cell culture. Along this line, recent emerging technologies have inserted, in the fluidic devices, commercially available trans-wells that can be easily removed from the dynamic circuit and analyzed with conventional techniques. Other strategies take advantages of integrated biosensors able to measure some metabolic parameters and soluble factors, over time, without the interruption of the dynamic culture [229,230].

Nevertheless, such platforms may also remain too simplistic, due to the lack of a realistic vasculature where immune cells can flow and transmigrate through the endothelial wall, reaching the tumor. Moreover, it is important to highlight that most of the studies have based their works on the use of long-term cultured cell lines, known to be remarkably different from primary tumors or immune cells. Future implementations should focus on the use of patient-derived tumors and immune cells with stromal components, eventually in multi-organ on-chip platforms, where the contribution of different organs to immunotherapies can be considered (Figure 2b).

The new in vitro 3D preclinical approaches may lead to personalized medicine predicting response to immunotherapies with greater patient benefits.

## Figures and Tables

**Figure 1 cancers-14-01013-f001:**
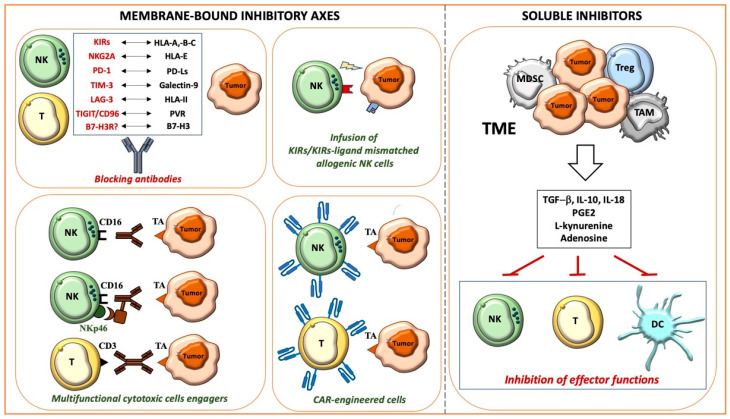
Main mechanisms allowing cancer immune evasion and therapeutic strategies.

**Figure 2 cancers-14-01013-f002:**
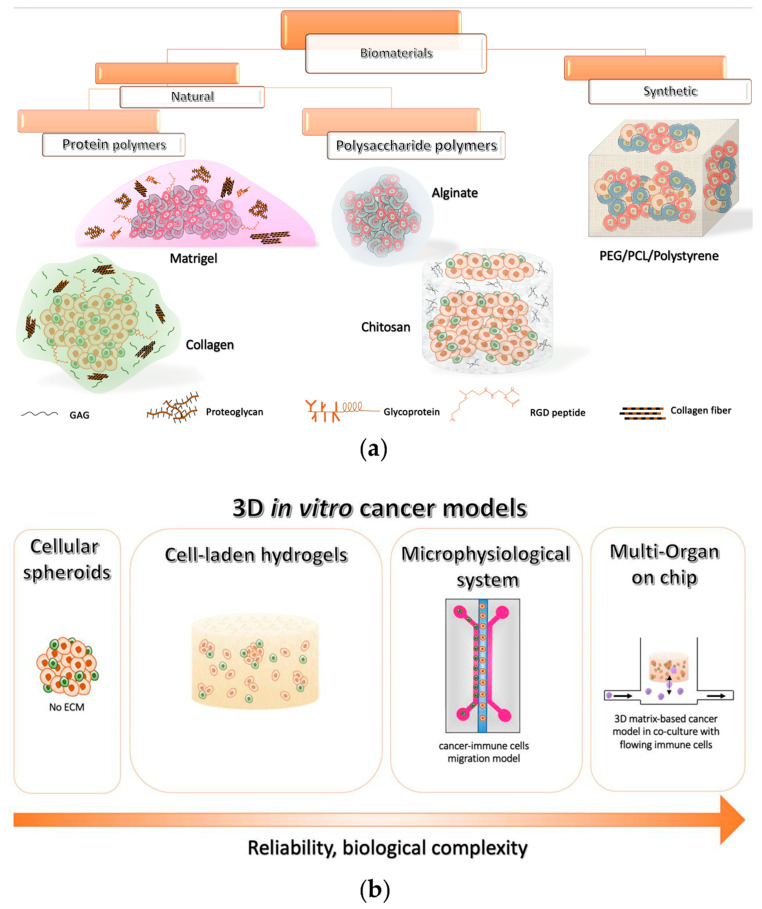
Overview of the current 3D in vitro cancer models (**a**) Classification of the most used polymeric biomaterials for 3D models and their schematic representation. (**b**) Evolution of 3D in vitro models for investigating cancer immunotherapies.

**Table 2 cancers-14-01013-t002:** Recent immune-on-chips for studying immunotherapeutic strategies against 3D cancer models.

	Microfluidic Device				
Key Immune Cell/Checkpoint Axis	3D Biomaterial	Cell Types	Method	Main Outcomes	Ref
T cells	Collagen	Human hepatocellular cell line (HepG2); TCR-T cells	Tumor aggregates in a central gel region with tumor-specific T cell receptors added in the adjacent channels	Chemotactic migration, effective cytotoxicity	[207,208]
	Gelatin Methacrylate	Human epithelial ovarian cancer cell line (SKOV3); CAR-T cells	Cancer cells in a central gel-filled region flanked with two channels where CAR-T cells reside	Enhanced cancer cell killing within a hypoxic TME	[209]
NK cells	Collagen	Breast cancer cell line (MCF7); NK-92 cell line; endothelial cells (HUVECs)	3D matrix containing cancer spheroids and NK-92 cells, provided with two lateral endothelialized channels	Chemotactic migration and penetration; cytotoxicity	[210]
		Cervical cancer cell line (Hela cells); NK-92 cell line	Injection molded plastic array culture (CACI-IMPACT) patterning gel regions including cancer cells; NK cells deposited over hydrogel structures	3D ECM remarkably reduces NK cell migration	[211]
Monocytes/ Macrophages	Collagen	Human hepatocellular cell line (HepG2); monocytes; TCR-T cells	Immune and cancer cells embedded in the central part of the microfluidic device, while T cells added in the channel	Immunosuppressive potential of monocytes via PDL/PDL-1 signaling	[212]
		Lung adenocarcinoma cell line (A549); TAMs; HUVECs	Tumor aggregates and TAMs included in 3D hydrogel, in co-culture with an endothelial monolayer in an adjacent channel	Tumor cell migration, epithelial–mesenchymal transition	[213]
		Mouse glioblastoma cell lines (GL261 and CT2A); macrophages; HUVECs	Hydrogel cancer and immune cells crossed by two inner vessels lined with HUVECs	M2-like macrophage polarization, angiogenesis promoted via TGF-β1 and IL-10	[214]
		Mouse macrophage cell line (Raw 164.7); human metastatic breast cancer cell line (MDA-MB-231); prostate cancer cell line (PC3); melanoma cell line (MDA-MB-435S); monocytes	Immune and cancer cells co-embedded in a central gel region	Macrophages promote cancer cell migration by upregulating MMP expression of tumor and secreting TNF-α and TGF-β1	[215]
		Human metastatic breast cancer cell line (MDA-MB-231); monocytes	Cancer and immune cells embedded in 3D hydrogel within an endothelial channel	Tumor cell extravasation promoted via monocyte-derived MMP9	[216]
		Human metastatic breast cancer cell line (MDA-MB-231); monocytic cell line (U937); TAMs	Two separated adjacent hydrogel channels containing cancer cells and monocytes or TAMs	Monocyte conversion to TAMs, promoted cancer cell invasion	[217]
		Pancreatic ductal adenocarcinoma cell line (CRL-1469); macrophages	Cancer and immune cells cultured in separated gel channels	Macrophage migration	[218]
	Collagen	Ovarian cancer cell line (OVCAR-3); neutrophils	Ovarian tumor spheroids embedded within hydrogel matrix with microfluidic channels carrying immune cells	Neutrophil extravasation, tumor cell migration	[219]
Neutrophils	Fibrin	Melanoma cancer cell line (A375-MA2), neutrophils; HUVECs	Cancer, immune, and endothelial cells co-embedded in a central hydrogel compartment	Increased tumor cell extravasation in an IL-8-dependent manner	[220]
DCs	Collagen	Colorectal cancer cell line (SW620); DCs	Cancer and immune cells cultured in a 3D chamber connected through microchannels to the immune compartments containing IFN-α-conditioned DCs	Crosstalk between dendritic and cancer cells	[221]
PD1/PDL-1	Collagen	Murine- and patient-derived melanoma cells; tumor-infiltrating lymphocytes	Organotypic tumor spheroids containing autologous immune cells embedded in a hydrogel	Effective response to PD-1 blockade treatment	[222]
ADCC	Collagen	Breast cancer cell line (BT474); CAFs; PBMCs; HUVECs	Central endothelial channel with two adjacent gel compartments including cancer cells, CAFs, and PBMCs	Trastuzumab antibody targeting the HER2 receptor promotes long cancer–immune interactions	[223]

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
