# Peer review of "Tumor Microenvironment and Hydrogel-Based 3D Cancer Models for In Vitro Testing Immunotherapies"

_cancers, 2022, doi:10.3390/cancers14041013_

Round 1
Reviewer 1 Report
- There are many reviews on the same subject on the last 2-3 years. The authors should clearly comment on the novelty of their work, compared to previously published reviews in the introduction and abstract.
- Most of the references in the introduction are not recent. I would suggest the authors to cite the important and most recent articles in this section.
- The introduction is not specific but rather vague: For example, authors wrote that “Experimental evidence proved that an altered ECM architecture and composition play a pivotal role in the tumor onset, progression, immune evasion, and sensitivity to immunotherapeutic approaches”. It would be more intriguing if the authors can provide specific information on how altered ECM structure/composition can induce tumor onset, progression or immune invasion. Most of the introduction can be rearranged in this way.
- The discussion of cancer immune-evasion mechanism is intriguing but can be shortened. The highlights of the paper is not to gain a knowledge on the invasive mechanisms rather how 3D models would benefit from identifying/studying existing or new mechanisms.
- A table with the use of different biomaterials and 3D models in studying cancer-immune cells interactions would highly increase the readability to the authors.
- In the conclusion and future direction, each problem/deficiency can be presented in subsections with a heading that must be accompanied with rationale and possible solutions.
- Other important missed point is the difficult of analyzing samples from these devices or models.
- Another missing point is the importance of investigating the synergy between ECM, chemotherapy, and immune cell’s function using 3D models. For example- a recent study highlights how chemotherapy induced changes in ECM may regulate immune cell’s functions (Cancer Res; 82(2) January 15, 2022). A discussion should be added to mimic such synergy between host-mediated mechanisms and ECM-remodeling.
- Only important, most relevant, and most recent (within the last 5 years) references should be cited. Although this is a review article, there is no need to cite more than 250 references. In fact, most of the references are back dated and this approach makes dilute the importance of relevant references.
Reviewer 2 Report
- This paper would benefit from more extensive use of figures to help explain and elucidate concepts.
- There is brief discussion about recapitulating key features of the ECM. I think it would be useful to move this concept up in the text and use it to guide the discussion of the various biomaterials. It would also be helpful to specify and delineate these functions so that you could then discuss how each material does or does not address these functions. Potentially you could also rank the relative importance of different functions in the development of tumor models to help guide readers in their selection of different materials.
- With the discussion of alginate, it is brought up that this material is not intrinsically cell adhesive. However, the work to modify alginate with cell adhesion peptides has been very widely reported in the literature and is quite easy to accomplish. This deserves greater coverage.
- Chitosan is NOT vegetable-derived. It is extracted from shrimp shells. You also need to include explanation of the relationship between chitin and chitosan and how this modification impacts mechanical properties and other material properties (like hydrophobicity). Perhaps it would be better to have your sections on naturally-derived matrices as Protein Materials and Polysaccharide Materials to avoid this confusion between animal and vegetable.
- It would be useful to include discussion of modification of PEG hydrogels to render them degradable by proteases such as MMPs.
- Table 1 does not add anything relative to the text. I suggest removing this.
